# Does Information Infrastructure Promote Low-Carbon Development? Evidence from the “Broadband China” Pilot Policy

**DOI:** 10.3390/ijerph20020962

**Published:** 2023-01-05

**Authors:** Hanjin Xie, Xi Tan, Jun Li, Shuang Qu, Chunmei Yang

**Affiliations:** 1School of Economic and Management, East China Jiaotong University, Nanchang 330013, China; 2School of Public Economics and Administration, Shanghai University of Finance and Economics, Shanghai 200437, China

**Keywords:** information infrastructure, low-carbon development, broadband China, carbon emission intensity

## Abstract

While information infrastructure has remarkably boosted global economic prosperity in the last several decades, how it propels low-carbon development has failed to draw enough attention. Based on panel data from 284 cities in China from 2006 to 2019, this study used the “Broadband China” pilot policy as an exogenous event to examine the impact of information infrastructure on carbon emission intensity. We found the following: (1) The “Broadband China” pilot policy significantly reduced carbon emission intensity, which held true in a series of robustness tests. (2) Promoting the development of the service sector, encouraging innovation activities, and fostering low-carbon lifestyles are the influential mechanisms by which information infrastructure reduced carbon emission intensity. (3) The population size, administration rank, marketization, industrialization, and informatization base significantly strengthened the reduction effect of information infrastructure on carbon emission intensity, while the disparity in human capital does not cause an evident difference. This paper’s findings reveal a counting path through which improving information infrastructure advances low-carbon and sustainable growth.

## 1. Introduction

Since the 18th National Congress, China’s informatization has improved at an unprecedented speed under the administration’s high priority. The “Communications Industry Statistical Bulletin 2020” shows that, in 2020, the annual revenue of the telecommunications business reached 1.36 trillion, the average monthly mobile Internet traffic per household surpassed 10 GB, and over 600,000 5 G base stations were built. As the expressway of the information age, information infrastructure allows broader applications of new-generation information technologies, such as 5 G, big data, and artificial intelligence, greatly enhancing the digital economy and informatization.

With booming informatization, China faces severe resource and environmental constraints. The growing greenhouse gas emissions hinder China’s sustainable path. To rein in carbon emissions, the Chinese government has established, to the international community, an ambitious carbon neutrality target that calls for peaking carbon emissions by 2030 and achieving net-zero carbon emissions by 2060. In the dilemma of development and emission reduction, the previous high-speed growth driven by massive energy consumption and CO_2_ emissions should rapidly give way to a more efficient and environmentally friendly one, relieving the pressure of economic growth on resources and the environment.

A number of studies have shown that information infrastructure, as network infrastructure, not only strongly promotes technological progress and productivity improvements but also has strong externalities, enabling it to produce positive spillover effects on a larger scale [1,2,3]. The influence of information infrastructure on socioeconomic development has attracted the attention of developed countries and developing countries in the world. The public sector has also made large investments in promoting the construction of information infrastructure, as the Chinese government has done. However, as far as sustainable development is concerned, the impact of developing information infrastructure seems uncertain. On the one hand, knowledge sharing and technology diffusion, which are beneficial for improving labour productivity and energy efficiency, can be accelerated by upgrading the information infrastructure [4,5]. In addition, compared to traditional manufacturing, industries nurtured and supported by informatization may be more independent of energy consumption. On the other hand, like public transportation infrastructure, the construction of information infrastructure drives a considerable investment inflow, expanding production and emissions [6,7]. At the same time, expanding information channels allows consumers to access a broader range of goods and services, thus stimulating consumption, which will undoubtedly raise carbon emissions [8,9].

The contradiction between the above two effects makes people doubt whether information infrastructure should be developed to a greater extent. Obviously, we would like to see that information infrastructure can effectively control the expansion of carbon emissions while shouldering the heavy responsibility of the economic growth engine. So what effect do the facts support? Can information infrastructure, the highway in the information age, propel low-carbon development? If so, what are the mechanisms?

At present, the existing empirical research does not clearly answer the above questions, but there are two types of studies related to it. One type of study has focused on the economic effects of information infrastructure: Roller and Waveman [1] examined the correlation between the telecommunication infrastructure level and the aggregate output based on data from 35 countries, and they found that although the two variables did not show a significant positive causality on average, the impact would be considerable when the telecommunication infrastructure reached a certain level. Duggal et al. [2] estimated a model based on a nonlinear production function, and the results show that information infrastructure investment was one of the largest contributing components of the American aggregate output from 1975 to 2001. Using data from 15 European countries, Koutroumpis [3] also found that doubling the broadband penetration rate would increase the total output by 2.6–3.8%. In addition, he confirmed the nonlinear relationship between information infrastructure and economic growth proposed by Roller and Waveman; that is, countries with high penetration rates have significantly greater elasticity coefficients of the broadband penetration rate to output. The subsequent research basically supports the above view [10,11,12], and all empirical evidence from different countries and regions reveals the important fact that information infrastructure investment is undoubtedly a powerful engine for economic growth.

In another type of study, researchers have discussed the relationship between other representative network infrastructures, such as transportation infrastructure, and environmental sustainability: Yang et al. [13] evaluated the improvement of environmental pollution in China caused by a high-speed railway opening, and the results based on the DID method show a SO_2_ emission reduction of about 7%. Dalkic et al. [14] included the travel mode and demand change in their model and found that high-speed rail travelling decreased 24.3 kt CO_2_ emissions. Chen and Whalley [15] investigated the environmental impact of another key transport infrastructure—urban rail transit. They found that the opening of the metro reduced carbon monoxide in Taipei by 5–15%.

To investigate whether information infrastructure impacts low-carbon development, the endogeneity of variables is also crucial. While most relevant empirical studies have used broadcasting penetration, telephone penetration, telephone service income, and cable length to measure the level of information infrastructure [1-3], these variables, which are also determined by the communication demand and information technology, are endogenous. Hence, using the variables above to conduct the examination might not correctly estimate the real effect of information infrastructure.

In 2013, in order to advance informatization, the Ministry of Industry and Information Technology (MIIT) and the Development and Reform Commission (DRC) of China implemented the “Broadband China” pilot policy, which requires pilot cities to formulate measures to improve the supplies of fixed broadband, mobile base stations, fibre-optic cables, and other information infrastructures based on the overall implementation plan and local conditions. From 2014 to 2016, MIIT and DRC gradually established three batches of cities including 120 cities as policy pilots.

The pilot policy provides an applicable natural experiment to estimate the welfare effects of information infrastructure, and using the DID method, a practical approach to addressing endogeneity, the estimated result can present a more reliable causality [16]. Therefore, using “Broadband China” as a natural experiment, we examined the treatment effect of the pilot policy on carbon emission intensity to reveal the causality link between information infrastructure and low-carbon growth. After the conclusion was confirmed by a series of robustness tests, we looked into the influential mechanisms of this effect. In addition to investigating the impact of information infrastructure on industrial development and innovation activities, we also highlighted its impact on the carbon emissions from residential energy use. Finally, based on empirical facts, we examined the heterogeneity of information infrastructure affecting low-carbon development so as to explore which factors play a key role in it.

Compared with existing research, this paper makes three main marginal contributions: Firstly, by verifying the causality between the information infrastructure supply and carbon emission intensity, we fill the gap in the field of relevant research and present empirical evidence that information infrastructure investment can promote low-carbon development. Secondly, we systematically put forward and tested the influential mechanisms of information infrastructure on low-carbon development, enriching the theoretical understanding in related research. Thirdly, we examined the heterogeneity characteristic of the impact of information infrastructure, providing a theoretical basis for informatization and sustainable development.

## 2. Policy Background and Theoretical Analysis

### 2.1. Policy Background

In August 2013, China’s State Council released the “‘Broadband China’ Strategy and Implementation Plan”, which aims to strengthen the strategic guidance and systematic deployment of information infrastructure construction. It also put forward targets for the information infrastructure to be built in China by 2020, such as fixed broadband subscribers, the broadband penetration rate, mobile subscribers, and the Internet access rate. The program is structured into three stages: the first stage, the comprehensive speed-up stage (by the end of 2013), focuses on fibre-optic and 3 G network construction, increasing broadband network access rates, and improving users’ online experiences. The second stage, the promotion and popularization stage (2014–2015), continuously promotes broadband network speed while expanding broadband network coverage and applications. The third stage, the upgrading stage (2016–2020), upgrades broadband networks and technology with the broadband network service quality, application, and broadband industry support capacity reaching the advanced international level.

In the same year, to implement the “Broadband China” strategy, MIIT and DRC jointly issued the “Management Measures for constructing ‘Broadband China’ Demonstration Cities/City Clusters” and launched a program for constructing “Broadband China” demonstration cities/city clusters. From 2014 to 2016, MIIT and DRC gradually established 120 cities/city clusters as pilot cities. Through the “Broadband China” demonstration pilot construction, the Chinese central authorities hoped to accumulate experience in promoting informatization and gain experience and references for the nationwide promotion of the “Broadband China” strategy, thus elevating regional informatization and promoting economic growth and transformation.

### 2.2. Theoretical Analysis

As an integral part of public infrastructure, information infrastructure can not only drive directly tied investments but also generate a wide range of positive external impacts, exerting far-reaching effects on individual decisions and macroeconomics. So, how exactly does information infrastructure reshape low-carbon development? To better understand this point, we performed a comprehensive analysis based on a systematic review of the existing literature from three aspects: promoting the service industry, encouraging innovation, and fostering low-carbon life.

#### 2.2.1. Information Infrastructure and Service Sector

Information infrastructure can contribute to service sector development through a direct effect and an indirect effect. The direct effect is the promotion of the output scale of IT-related industries. As the basis for the application of information technology, increasing the information infrastructure enhances the expected output of its IT-related industries, which in turn draws investment and high-quality labour, stimulating the expansion of scale production. Moreover, the improvement of the information infrastructure can extend the application scenario and scope of information technology and enrich products and services based on information technology, spurring the final and intermediate demand for related products and services.

The indirect effect refers to the role of information infrastructure in reducing transaction costs. Information infrastructure can break the geographical barriers of transactions, reducing the time costs and transportation costs in transactions. In addition, information infrastructure can expand the boundaries of information searches, reducing the cost of information searching and mitigating the potential risks caused by information asymmetry [17]. Compared with the agricultural and industrial sectors, the goods produced by the service sector tend to be more differentiated and individualized, leading to higher information search costs and asymmetric risks. Thus, the service sector can benefit more from the reduction in transaction costs caused by an increase in information infrastructure. According to Baumol [18], due to lower transaction costs, the service sector can reduce goods prices to a greater extent, thus attracting an inflow of production factors and increasing the output share of the service sector in the economy.

The causality between the industrial structure and low-carbon development has been discussed extensively. One of the most classical discussions is the “structural effect” from the “environmental Kuznets hypothesis” proposed by Grossman and Krueger [19]. It states that when the dominant sector of the economy shifts from a highly polluting and energy-intensive heavy industry to a low-polluting service and knowledge-intensive industry sector, the input structure changes, and the emissions per output decreases. In most cases, production in the service sector requires lower-energy inputs relative to the industrial sector [20]. In addition, productive service industries in the service sector can provide productive services such as finance, technology, marketing, management, and transportation to other industrial sectors, increasing the output efficiency of the economy [21].

#### 2.2.2. Information Infrastructure and Innovation

The idea that information infrastructure encourages innovation has been extensively validated [22,23,24]. The views of existing studies on how information infrastructure promotes innovation can be summarized in two ways. One is to reduce the innovation cost. Hendriks points out that information communication technology (ICT) can break spatial and temporal barriers between knowledge producers, mitigating the friction in knowledge sharing and innovation collaboration and improving the innovation output and quality [24]. In addition, when the information infrastructure has a wider coverage of users, the marginal cost of knowledge acquisition will rapidly decline, while the marginal benefit gradually goes up. Moreover, accelerating the speed of information transmission helps to alleviate information asymmetry and optimize the allocation of innovation resources [17].

What also counts is promoting knowledge accumulation. The new growth theory casts human capital accumulation as the core driver of technological progress, which in turn determines economic growth [25]. Information infrastructure can speed up information interactions, rapidly enhancing the knowledge stock in the whole economy. In addition, upgrading and popularizing the information infrastructure allow knowledge exchange and technology diffusion across regions, from which a wider range of people reap the benefit, promoting technical advancement and the level of human capital [4,5,26,27].

According to Porter, innovations by firms to upgrade production technologies and equipment can improve their productivity and remould the input structure in the long run [28]. Although innovation in production technology may also induce production expansion and cause a rebound effect in total emissions [29], energy consumption and emissions per output should theoretically fall.

#### 2.2.3. Information Infrastructure and Low-Carbon Lifestyle

Information infrastructure can reshape people’s consumption habits and reduce carbon emission intensity in the residential sector, promoting a low-carbon lifestyle. First, improving the information infrastructure allows information technology to be applied in extended scopes and scenarios, making online shopping, distance education, and online offices possible. When consumers are more prone to choose online goods and services, offline products will be crowded out, and the demand for transportation will drop accordingly, resulting in lower physical materials and energy consumption per output. Second, online businesses are less spatially constrained than offline businesses: with the expanded coverage of the information infrastructure, enterprises can use information networks to access a larger consumer market, pursuing a large-scale economy. Finally, information infrastructure broadens consumer access to information, allowing consumers to purchase appropriate amounts of goods and services and avoid waste. Consumers can obtain information about products’ green labels, energy intensity, lifetime, and other information, increasing the likelihood that they will purchase clean and energy-efficient products, with all other things being equal. In addition, by jumping on the information network bandwagon, the idea of environmental protection and ecological culture can become more widespread, which helps to improve public environmental literacy and drive low-carbon consumption [30,31].

Existing studies worldwide have shown that residential consumption is one of the vital sources of total carbon emissions. A study by Kim [32] on carbon emissions from residential consumption in Korea from 1985 to 1995 also showed that residential energy consumption and the demand for energy-intensive products are the most critical emission factors. In the European Union, households have surpassed industries as the largest source of energy consumption since the 1990s [33]. Considering the residential sector’s domination of the total carbon emissions, a low-carbon lifestyle should strongly drive the reduction in the overall emission intensity, propelling low-carbon development.

### 2.3. Hypotheses

Based on the above analysis, we can point out that information infrastructure has an extensive and far-reaching welfare effect on low-carbon development, which holds true from the macroeconomic structure to microconsumers. Overall, we argued that improving the information infrastructure can reduce the carbon emission intensity, inducing low-carbon growth. While there is a potential rebound effect of information infrastructure, expanding the scale of production and consumption, its role in promoting energy efficiency, improving technology, and greening lifestyles remains significant, and the latter is obviously more conducive to controlling increasing carbon emissions and pursuing low-carbon growth in the long run. Based on this, we propose the following four hypotheses:

**Hypothesis** **1** **(H1).***Information infrastructure can reduce carbon emission intensity, inducing low-carbon development*.

**Hypothesis** **2** **(H2).***Information infrastructure can induce low-carbon development by promoting the service industry*.

**Hypothesis** **3** **(H3).***Information infrastructure can induce low-carbon development by encouraging innovation*.

**Hypothesis** **4** **(H4).***Information infrastructure can induce low-carbon development by fostering a low-carbon lifestyle*.

## 3. Empirical Strategy

### 3.1. Baseline Model

We considered the “Broadband China” pilot policy as a quasi-natural experiment and used a difference-in-difference model to examine the impact of the “Broadband China” pilot policy on carbon emission intensity so as to investigate the causal link between information infrastructure construction and low-carbon development:(1)$$CEI_{it}=\alpha +\beta BBC_{it}+{X_{it}}^{\prime }\varphi +\mu_{i}+\gamma_{t}+\varepsilon_{it}$$
where *i* denotes the city, and *t* denotes the year. *CEI_it_* is the explained variable, carbon emission intensity. *BBC_it_* is the core explanatory variable, which equals 1 if city *i* is a “Broadband China” pilot in year *t*, and 0 if the opposite. *X_it_*’ denotes a group of control variables, *μ_i_* is the city fixed effect, *γ_t_* is the year fixed effect, and *ε_it_* is the random error. *β* is the coefficient of interest, which represents the changes in carbon emission intensity caused by the “Broadband China” pilot policy.

### 3.2. Variables

#### 3.2.1. Carbon Emission Intensity

We used carbon emission intensity (*CEI*), i.e., carbon emissions per GDP, in this study to measure low-carbon development. Firstly, referring to Glaeser and Kahn [34], we estimated the quantity of carbon dioxide emissions from various types of energy consumption in 284 cities from 2006 to 2019. Then, the total emissions were obtained by summing up the emissions from various types of energy consumption. Finally, the carbon emissions per GDP were calculated to obtain the carbon emission intensity:(2)$$CE_{it}=E_{it}\times EF_{\alpha ,t}+\sum_{\beta =1}^{2}G_{\beta ,it}\times GF_{\beta }+\sum_{\gamma =1}^{2}(H_{\gamma ,it}/(\theta \times LHV))\times CF$$
(3)$${CEI}_{it}=\frac{{CEI}_{it}}{{GDP}_{it}}$$
where *i* denotes the city, and *t* denotes the year. *CE_it_* is the total carbon emissions. *E_it_* is the electricity consumption. *EF_α,t_* denotes the carbon emission factors for the regions North China, Northeast China, East China, Central China, Northwest China, South China, and Hainan Province, respectively, as *α* varies from 1 to 7 (as the Hainan provincial grid was integrated into the Southern Grid in 2011, we adopted the same emission factor for cities in Hainan Province and South China from 2011 to 2019). *G_β,it_* and *GF_β_* denote the consumption and the carbon emission factor of coal and natural gas when *β* equals 1 (since the consumption amount of coal gas and natural gas are counted together in the data source and the consumption of coal gas accounts for a very small amount during the sample period, we applied the emission factor of natural gas to calculate emissions from coal gas consumption), and they denote those of LPG (liquefied petroleum gas) when *β* equals 2. *H_γ,it_* denotes the heating supply of steam and hot water when *γ* equals 1 and 2, respectively. *LHV* is the low heating value of raw coal, which equals 20,908 kJ/kg, and *θ* is the thermal efficiency, which equals 0.7 (According to “GB/T15317-2009 Energy Conservation Monitoring of Coal-fired Industrial Boilers”). *CF* is the raw coal carbon emission factor, which equals 2.53 kgCO_2_/kg, calculated by multiplying the standard coal carbon emission factor by the conversion ratio of raw coal to standard coal, which equals 0.7143.

#### 3.2.2. “Broadband China” Pilot Policy

The dummy variable *BBC*, i.e., “Broadband China” pilot policy, is used to measure the increasing supply of information infrastructure. There are three batches of pilots, which include 120 cities that implemented the “Broadband China” pilot policy in 2014, 2015, and 2016, respectively. After excluding the pilot cities with missing data and county-level pilot cities that implemented, the number of pilot cities, i.e., the number of treated cities, is 108 in this paper.

#### 3.2.3. Control Variables

Referring to existing studies, the following variables were selected to control the known factors correlated with low-carbon development: GDP per capita (*Pgdp*) and the square of per capita GDP (*Pgdp2*) [19]; total population (*Pop*) [35]; the structure of public expenditure (*Pub*) [36]; transportation infrastructure (*Trans*) [37]; human capital (*Hum*) [38]; and foreign direct investment per GDP (*Fdi*) [39]. In addition, the climate is also an important determinant of energy consumption. Summer and winter are the peak periods of energy use in a year, so average temperatures in summer and winter (*Sum* and *Win*) are also included in the model. The definitions and descriptive statistics of the above variables are given in Table 1.

### 3.3. Data Source

In carbon emission accounting, the data on electricity consumption and coal and natural gas consumption were obtained from the “China Urban Statistical Yearbook”. The heating supply of steam and hot water was obtained from the “Statistical Yearbook of Urban Construction”. The carbon emission factors of electricity were obtained from the “Emission Factors of China Regional Power Grid Baseline for Emission Reduction Project”. The carbon emission factors and the conversion factors of fossil fuels were gathered from the “IPCC National Greenhouse Gas Emission Inventory Guidelines 2006”. The pilot name list for “Broadband China” was collected from the website of The Central Government of China. The population data were obtained from the annual government work reports of cities, and the other control variables were obtained from the “China Urban Statistical Yearbook”. The temperature data were obtained from the website of The National Meteorological Administration of China.

## 4. Results

### 4.1. Baseline Regression

Based on Equation (1), we examined the impact of the “Broadband China” pilot policy on carbon emission intensity. The results of the baseline regressions are reported in Table 2, where control variables are not included in column 1 but included in column 2, and the city cluster standard error is listed in column 3. In the three regressions, the estimated coefficient of *BBC* is significantly negative, at least at 5%, indicating that the “Broadband China” pilot policy statistically reduced carbon emission intensity. On average, due to the “Broadband China” pilot policy, the increased information infrastructure supply led to an 11.4% decrease in carbon emission intensity.

### 4.2. Endogeneity

The nonrandom distribution of the “Broadband China” pilots may lead to an estimation bias in the baseline regression. Specifically, central authorities have the incentive to choose cities where infrastructure construction is much easier and cities with larger population scales when identifying the pilot cities. On the one hand, by choosing the former, the government can effectively control costs, avoiding inefficient public expenditure. On the other hand, by choosing the latter, the effectiveness of the policy can be improved due to the scale economy.

To solve this, we selected the relief degree of land surface (*Rdls*) [40] as an instrumental variable to conduct a Heckman two-step estimation [41]. *Rdls*, measuring the complexity of topography changes in a given region, is an appropriate instrumental variable for this paper. First, *Rdls*, as a geographical indicator, is strongly exogenous. Geographic factors have been good exogenous shocks in the economic field for a long time [42]. Their attributes mean that it is difficult to form a direct relationship with carbon emission intensity, and they are basically unaffected by urban economic activities [43]. Second, *Rdls* significantly and negatively correlates with the infrastructure supply in a region. This is because, in general, the more undulating the terrain and the harsher the geographical conditions, the more difficult it is to lay fibre-optic cables, base stations, and other infrastructure. In addition, the distribution of China’s economic geography is such that industries and populations are mainly concentrated in plains and gently sloping areas, while areas with poor topographic conditions are more likely to have poor economic performance and small populations [35]. Therefore, based on the cost–benefit trade-off, the central government tends to select cities with a lower relief degree of land surface as pilots.

Column 4 of Table 2 reports the results of the two-step regression. The estimated coefficient of *Rdls*, reported at the bottom of the table, is significantly negative in the first-step regression. The Cragg–Donald Wald F-statistic identifying the weak instrumental variable is 72.71, higher than the critical value of 16.38 for a 10% maximal IV size, indicating that the instrumental variable *Rdls* was appropriately selected. Furthermore, the result of the second-step regression shows that *BBC* is still significantly negative at 1%, which is consistent with the baseline regression results.

### 4.3. Pretreatment Trend and Annual Treatment Effects

The DID model should be adopted with the parallel trend assumption being satisfied: i.e., the treated and untreated cities should have the same pretreatment trend in carbon emission intensity. Referring to Alder et al. [44], we used the event study method to test the parallel trend assumption:(4)$$CEI_{it}=\alpha +\sum_{k=-9}^{6}\beta_{k}\times DT_{ik}+{X_{it}}^{\prime }\varphi +\mu_{i}+\gamma_{t}+\varepsilon_{it}$$
where *∑DT_ik_* is a set of dummy variables, and *DT_ik_* equals 1 in the k-th year that city *i* has implemented the “Broadband China” pilot policy and is 0 otherwise. *β_k_* represents a group of estimated coefficients of interest. As city fixed effects control for the inherent differences in carbon intensity between the treated and untreated cities, *β_k_* measures the differential changes in carbon intensity in the *k*-th year of the “Broadband China” pilot policy. For *k* < 0, *β_k_* is not significant, indicating no significant difference in the pretreatment trend between the two subclasses, and thus, the parallel trend assumption is verified.

The {*β_k_*} values estimated according to Equation (5) are reported in Figure 1. It can be seen that the coefficients before the treatment {*β_k_*, *k* < 0} are all insignificant at 10%, and overall, the trend is smooth. This indicates that the parallel trend assumption is satisfied.

The absolute values of the coefficients tend to increase roughly after the treatment. The coefficients are insignificant in the first three years and start to be significant in the fourth year. This result indicates that the “Broadband China” pilot policy has a treatment effect on carbon emission intensity that increases yearly. The first three insignificant coefficients may be related to the fact that the increased supply of information infrastructure might still be under construction in the early stage of the “Broadband China” pilot policy.

### 4.4. Placebo Test

To exclude interference from unobservable factors on the estimation results, we conducted a placebo test with 1000 random samples. To improve the accuracy of the test, we constructed both a fictitious policy pilot and fictitious policy time so that the policy generated by the placebo test can cover a wider range of possibilities.

Specifically, first, any year between 2006 and 2019 was randomly selected for each city as the year of policy implementation. Then, keeping the number of cities in the treated subclass consistent, 108 cities were randomly selected as the new treated subclass, while the rest were assigned to the untreated subclass. Next, the baseline model was regressed based on the fictitious policy. Finally, repeating the above steps 1000 times, 1000 fictitious estimated coefficients {*β^random^*} and *p*-values {*p-value*} were obtained.

Figure 2 shows the kernel density distribution of {*β^random^*}, which is basically normally distributed, and the baseline regression coefficient of −0.114 (reference line in Figure 2) is located at the edge of the distribution. By looking up the empirical cumulative distribution function of {*β^random^*}, it can be found that the baseline regression coefficient of −0.114 lies in the interval (−0.1146, −0.1123) in the distribution. Correspondingly, the cumulative probability lies in (0.6%, 0.7%), indicating that the baseline regression coefficient is significantly different from the distribution of {*β^random^*}, at least at 5%.

Figure 3 shows the scatter plot of {*p-value*} against {*β^random^*}. It can be seen that the majority of the coefficient’s *p*-values (891 of 1000 scatters) lie above the reference line y = 0.1, i.e., insignificant. By viewing the descriptive statistics of the {*p-value*}, it is also found that there are only 12 observations below the *p*-value of the estimated coefficient in the baseline regression (−0.1139). The above results suggest that the placebo test is as expected and the baseline regression results are reliable.

### 4.5. Robustness Test

#### 4.5.1. Spatial Spillover

The previous conclusion is based on the assumption that the impact of the “Broadband China” pilot policy on carbon emission intensity is only effective in the pilot cities, and there is no spatial spillover. However, this assumption may be difficult to meet. Due to trade, the division of labour, and migration, there are always strong or weak economic linkages between cities. In most cases, these linkages are stronger between nearby cities according to the first law of geography [45].

Consider that the policy has a positive spillover effect on all neighbouring untreated cities and has zero spillover effect on neighbouring treated cities or the policy spillovers between treated cities exactly cancel each other out. In this case, even if the local effect of the policy is not significant, it seems that the carbon intensity of the treated cities still decreases compared to the untreated cities, and the previous conclusion is absolutely spurious.

To address this, we describe the spatial spillover of the policy with two spatial weighting matrices, contiguity and inverse distance. We include the policy spatial lag in the regression to control for the effect of policy spatial spillover. The regression results in columns 1–2 of Table 3 show a positive policy spatial spillover, while only the spatial spillover with the contiguity weighting matrix is significant. More importantly, after controlling for policy spatial spillovers, the coefficients of *BBC* are still significantly negative and consistent with the results of the baseline regression, indicating that the local effect of the policy still exists.

#### 4.5.2. Pre-Existing Trends

The baseline model with two-way fixed effects can control for inherent city characteristics and common trends across all cities. However, specific inherent characteristics of cities may also impact carbon intensity over time. For a long time, industrial structure, energy resource endowment, and heating practices have differed between cities in northern and southern China. Moreover, these differences not only led to an inherent gap in carbon emission intensity between northern and southern cities but also, due to path dependency, dominated the development mode of cities, which is correlated with the carbon emission reduction potential.

Another scenario is that China’s provincial capitals, as regional economic centres, have a clear advantage over other cities in terms of infrastructure and public service. This advantage will gradually snowball through labour migration, capital flows, and even policy tilt, improving the carbon emission efficiency of provincial capitals in the long run.

To control for the long-run effect of these location factors, we included the interaction of the north dummy variable with the linear time trend (*Nor_T*) and the interaction of the provincial capital dummy variable with the linear time trend (*Cap_T*) in the baseline regression. The results (Table 3, columns 3–4) show that the estimated coefficients of *BBC* decrease but are still significantly negative after controlling for the long-run effect of location factors.

#### 4.5.3. Carbon Emission Per Capita

To test the sensitivity of the previous findings to the way that low-carbon development is measured, we used carbon emissions per capita to regress according to Equation (1). The results (Table 4, columns 1–2) show that the treatment effect of the policy remains significantly negative, at least at 5%, regardless of the inclusion of control variables.

#### 4.5.4. PSM-DID

The differences in various characteristics between the treated and untreated cities may cause an estimation bias, so we screened the sample by using Propensity Score Matching (PSM). First, to avoid too small a sample size after matching and to ensure matching accuracy, we selected *Pgdp*, *Pop*, *Sum*, and *Win*, which are robustly significant in the baseline regression (Table 2, columns 1–3) as matching variables. Second, to address the problem of having no treatment variable before the policy implementation (2014), we chose *BBC_i_*_,2014_ as the treatment variable for the sample from 2006 to 2013. Finally, the treated and untreated cities were matched yearly according to the 1:2 nearest-neighbour matching method.

Column 3 of Table 4 shows the regression results based on the sample on a common support. Column 4 of Table 4 shows the result using frequency-weighted regression. Still, the coefficients of *BBC* remain consistent with those estimated from the baseline regression.

#### 4.5.5. Dynamic Model

Due to the continuity of economic development, both the aggregate output and energy use are unlikely to change to a considerable extent in a short period of time, which means the explained variable *CEI* in this paper may have autocorrelation. More importantly, this issue may interfere with the estimated result of the core explanatory variable (BBC). To address this, we included the lags of *CEI* in the explanatory variables to construct a dynamic panel model and used Difference GMM and System GMM methods to estimate it.

The results show that after controlling the first-, second-, and third-order lags of *CEI*, the coefficients of the core explanatory variable *BBC* decreases, but they are still significant at least at the 5% level (Table 5). In addition, in general, the first- and second-order lags of CEI have a significant impact on CEI, but for the third-order lag, this effect disappears, indicating that the change in carbon emission intensity is a dynamic process.

## 5. Influential Mechanisms

We examined the hypothesis that information infrastructure can promote low-carbon development in the previous sections. Further, how exactly did information infrastructure achieve carbon emission reductions? Based on the analysis in Section 2.2, this paper will verify the influential mechanism of information infrastructure on low-carbon development in three aspects: promoting the service industry, encouraging innovation, and encouraging a low-carbon lifestyle.

### 5.1. Information Infrastructure and Service Industry

An essential manifestation of the development of the service industry is the increase in the output share of services. In addition, a productive service industry can empower the industrial sector and improve industrial energy efficiency. We measured the service industry development by the output share of the service industry (*Sid1*) and the employment proportion of the productive service industry to the service industry (*Sid2*) (transportation, storage, post and telecommunications industry, finance industry, and information and computer software industry were selected as productive service industries).

The results of the regressions with *Sid1* and *Sid2* as explained variables show that the coefficients of *BBC* are significantly positive (Table 6, columns 1–2, at least at 10%). This indicates that the “Broadband China” pilot policy has significantly contributed to the increase in the output share of the service industry and promoted the upgrade of the service industry. Thus, Hypothesis 2 of this paper is verified.

Generally, the dependence of economic growth on energy consumption decreases as the output share of services increases. At the same time, the expansion of the productive service sector can provide the industrial sector with services in financing, transportation, CNC communication, marketing, etc., helping the industrial sector to improve its efficiency in production, management, and marketing, thus increasing the energy use efficiency.

### 5.2. Information Infrastructure and Innovation

Innovation is the core dynamic of industries to improve product quality and production efficiency. Technical improvements and investments in cleaner equipment can effectively increase outputs per carbon emission. To verify the influential mechanism whereby information infrastructure promotes low-carbon development by encouraging innovation, we used the patent data from cities to measure the outputs of innovation activities.

Specifically, we estimated the treatment effects of the “Broadband China” pilot policy on per capita patents (*Inn1*), per capita invention patents (*Inn2*), and per capita utility model patents (*Inn3*). The results show that *BBC* values are all significantly positive (Table 6, columns 3–5, at least at 5%), indicating that “Broadband China” indeed encouraged innovation activities in cities, thus confirming Hypothesis 3. The rise in the outputs of innovation activities implies an increase in the application of highly productive technology and equipment and then the outputs per energy consumption: carbon emission intensity, in other words, grows with the improvement in productivity.

### 5.3. Information Infrastructure and Low-Carbon Lifestyle

To verify the influential mechanism whereby information infrastructure promotes low-carbon development by promoting a low-carbon lifestyle, we conducted a series of regressions with the carbon emission intensity of different types of residential direct energy consumption as explained variables. These indicators were used to measure a low-carbon lifestyle for two reasons: first, direct energy consumption in the residential sector is an essential source of carbon emissions in the residential sector, and energy-saving is also an influential sign of the greening of lifestyles. Second, by examining the change in the intensity of carbon emissions from residential transportation energy consumption, we can determine whether the information infrastructure has promoted an online lifestyle.

Initially, we adopted the Emission-Factor method to calculate the residential sector’s carbon emissions originating from four direct energy consumption items: electricity, gas, heating, and transportation (gas consumption includes coal and natural gas and liquefied petroleum gas consumption, and transportation includes travel by private cars, buses, and cabs):(5)$$\begin{array}{l} ECE_{it}=E_{it}\times E_{\alpha ,it};\;\;GCE_{it}=\sum_{\beta =1}^{2}F_{\beta ,it}\times FC;\\ HCE_{it}=HA_{it}\times CA_{i}\times CF;\;\;TCE_{it}=\sum_{\gamma =1}^{3}V_{\gamma ,it}\times L_{\gamma }\times G_{\gamma } \end{array}$$
(6)$$RCEI_{it}=\frac{ECE_{it}+GCE_{it}+HCE_{it}+TCE_{it}}{GDP_{it}}$$
(7)$$ECEI_{it}=\frac{ECE_{it}}{GDP_{it}};GCEI_{it}=\frac{GCE_{it}}{GDP_{it}};HCEI_{it}=\frac{HCE_{it}}{GDP_{it}};TCEI_{it}=\frac{TCE_{it}}{GDP_{it}}$$
where *i* denotes the city; *t* denotes the year; and *ECE_it_*, *GCE_it_*, *HCE_it_*, and *TCE_it_* denote carbon emissions from electricity consumption, gas consumption, heating energy consumption, and transportation energy consumption, respectively. The factors and variables in the calculation of electricity and gas carbon emissions (*ECE* and *GCE*) are in line with those in Section 3.2. Heating carbon emissions (*HCE*) are calculated by using the total residential heating area (*HA_it_*), coal consumption per heating area (*CA_i_*), and the standard coal emission factor (*CF*). Carbon emissions from transportation energy consumption are calculated by using the quantity, the average annual mileage, and the fuel coefficients of each type of vehicle. Referring to Zhang et al. [46], private cars (γ = 1), cabs (γ = 2), and buses (γ = 3) are included in the calculation, and the factors are as follows: *L*_1_ = 70,000 km, *L*_2_ = 12,000 km, *L*_3_ = 20,000 km, *G*_1_ = 0.32 L/km, *G*_2_ = 0.1 L/km, and *G*_3_ = 0.1 L/km. (The data on residential electricity consumption, residential coal and natural gas consumption, residential liquefied petroleum gas consumption, and the number of cabs and buses were obtained from the “China Urban Statistical Yearbook”. The residential heating area was collected from the “China Urban Construction Statistical Yearbook”. The data on the number of private cars were gathered from the annual statistical bulletin of each city. Coal consumption per heating area data were gathered from the official website of China’s Ministry of Housing and Urban-Rural Construction).

The carbon emission intensity of residential total direct energy consumption (*RCEI*) is defined in Equation (6), and the carbon emission intensity of residential electricity/gas/heating energy/transportation energy consumption (*ECEI*; *GCEI*; *HCEI*; *TCEI*) is defined in Equation (7).

Table 7 presents the regression results of the impact of information infrastructure on low-carbon lifestyles. Overall, the “Broadband China” pilot policy has a significant reduction effect on the residential carbon emission intensity from direct energy consumption, and the effect holds for *RCEI*, *ECEI, GCEI,* and *TCEI* but not *HCEI.* In general, this implies that increasing the information infrastructure can decrease the carbon emission intensity of the residential sector, fostering a low-carbon life.

## 6. Heterogeneity

The effect of information infrastructure on carbon emission intensity may be heterogeneous due to the fundamental differences in economic development stages and factor endowments. For cities in the early stage of industrialization, even if the government increases the information infrastructure supply and the transaction cost of the service sector falls, the return of the industrial sector may still be much higher than that of the service sector. Hence, the dominant sector of the economy should still be the industrial sector, and the proportion of service outputs will not increase significantly.

Therefore, in order to comprehensively evaluate the impact of information infrastructure on low-carbon development and provide low-carbon development with practical insights, we conducted grouped regressions to explore the heterogeneity of information infrastructure effects on carbon emission intensity. To be specific, we divided treated cities into high-level and low-level groups according to the median or mean of the grouping variables, while the untreated cities remained unchanged. Then, the two groups were combined with untreated cities separately to conduct a difference-in-difference regression. With the same untreated subclass, such grouped regression can compare the treatment effects in the high-level group with those in the low-level group. Additionally, to avoid potential endogeneity and to ensure the accuracy and relevance of the grouping, the grouping variables were selected from cross-sectional data from 2013, the year before implementing the pilot policy, “Broadband China”.

### 6.1. Population Size and Administration Rank

Firstly, we divided the treated cities into two groups according to the median permanent resident population to examine the impact of the population size heterogeneity on the treatment effect. Then, by splitting the cities into provincial capital cities and other cities, we examined the impact of the heterogeneity in the administration rank on the treatment effect.

The regression results in Table 8 show that the treatment effects of the pilot policy are markedly stronger in provincial capital cities and cities with larger population sizes. In contrast, the treatment effects in non-provincial cities and cities with smaller population sizes are significantly weaker or even insignificant. A larger population implies that the increased supply of information infrastructure allows more people to benefit from the information network, leading to a greater scale effect. With a higher administration rank, the advantage of cities in the allocation of innovation factors is more prominent, generating more innovation outputs.

### 6.2. Human Capital and Informatization Base

High levels of human capital are imperative for innovation. More innovation outputs can be yielded through knowledge sharing and innovation cooperation with higher levels of human capital in a region. However, by shattering spatial restrictions, information infrastructure vastly expands the geographical scope of knowledge sharing and innovation cooperation, making knowledge exchange and innovation cooperation in distant regions much more feasible. Hence, the human capital’s impact on the effect of information infrastructure on carbon emission intensity may not be as strong as expected.

Typically, advanced information appliances, governance, technology, and service will allow information infrastructure to play a far-reaching role in promoting low-carbon development.

We measured human capital and the informatization base using the number of college students per 10,000 population and the number of employees in the information industry per 10,000 population, respectively. Table 9 reports the regression results based on different levels of human capital and informatization base. The results suggest that the difference in human capital did not significantly distinguish the cities from each other on the basis of carbon emission intensity, while the heterogeneity in the informatization base did make a difference. This may reveal that information infrastructure can cover the gap in human capital by propelling knowledge spillover and collaborative innovation across regions, while the welfare effects of public investment in information infrastructure are still dominated by the Matthew Effect.

### 6.3. Marketization and Industrialization

Information infrastructure optimizes innovation factor allocation by broadening information channels and moderating information asymmetry, and market orientation can promote the efficiency of factor allocation. The industrial sector, the foundation of information-based development, provides material support for information infrastructure construction. Therefore, the information infrastructure should have a stronger effect on carbon emission intensity with a higher level of marketization and industrialization.

We used the Provincial Marketization Index [47] and the output proportion of the industrial sector and service sector to measure marketization and industrialization, respectively. The regression results are shown in Table 10. Apparently, the treatment effects of the “Broadband China” pilot policy were significantly differentiated from each other under various degrees of marketization and industrialization. The higher the degrees of marketization and industrialization, the stronger the reduction in carbon emission intensity.

## 7. Conclusions

The increased supply of information infrastructure has contributed greatly to global economic prosperity in the past few decades. However, insufficient attention has been paid to the path through which information infrastructure boosts energy savings, emission reduction, and green development. We have examined the impact and mechanisms of information infrastructure on low-carbon development using the “Broadband China” pilot policy as a natural experiment. The results suggest that information infrastructure significantly lowers carbon emission intensity. These results, reconfirmed by IV regression and a placebo test, still held true in robustness tests, including when controlling for spatial spillover and the pre-existing trend, changing the measurement of the explained variable, and performing PSM-DID regression. The annual treatment effects based on an event study suggest that raising the information infrastructure supply, driven by the “Broadband China” pilot policy, does not work immediately but gradually improves as time goes on.

The mechanism test shows that information infrastructure reduced carbon emission intensity by fostering service industry development, innovation, and a low-carbon lifestyle. First, information infrastructure promoted the development of the service sector, particularly the productive service sector. Second, information infrastructure remarkably encouraged innovation activities, leading to an increase in innovation outputs, which are measured by patents per capita. Finally, information infrastructure cultivated a low-carbon lifestyle, reducing the carbon emission intensity of the residential sector. Further, the heterogeneity test found that population size, the administration rank, marketization, industrialization, and the informatization base significantly strengthened the reduction effect of information infrastructure on carbon emission intensity, while the disparity in human capital did not cause an evident difference.

By examining the policy effect of the “Broadband China” strategy pilot on carbon emission intensity, we show a counting path, enhancing information infrastructure for sustainable and low-carbon development. Under economic depression and COVID-19, information infrastructure will play a greater role in recovering the global economy. Countries or regions should attach unprecedented importance to information infrastructure, supporting the global application of 5 G, artificial intelligence, and big data, and pursuing a sustainable digital economy.

Differing from existing studies, we have focused not only on the impact of information infrastructure on low-carbon production but also on its role in low-carbon life. In this way, we comprehensively understand the sustainable development impetus brought by information infrastructure investment. However, generally, our research was designed to reveal the causal link between information infrastructure investment and low-carbon growth, and empirical examinations were conducted at the city level. At the micro level, the role of information infrastructure in shaping the environmental performance of economic units, such as individuals and enterprises, may be omitted.

To further investigate this field, researchers could extend this work to the relationship between information infrastructure and household consumption carbon emissions. Another potential opportunity is exploring the effects of information infrastructure on enterprises’ environmental performance. From a micro perspective, researchers can examine how information infrastructure influences consumer and production decisions, which are generally less studied. These kinds of research can provide more convincing and comprehensive empirical evidence about the environmental consequences of informatization, and they also help understand sustainable informatization. In the context of growing carbon emissions, continuing to explore the micro motivation factors of low-carbon life or low-carbon production in the information era is of critical significance for countries and regions to rise to the challenges that climate change brings.

## Figures and Tables

**Figure 1 ijerph-20-00962-f001:**
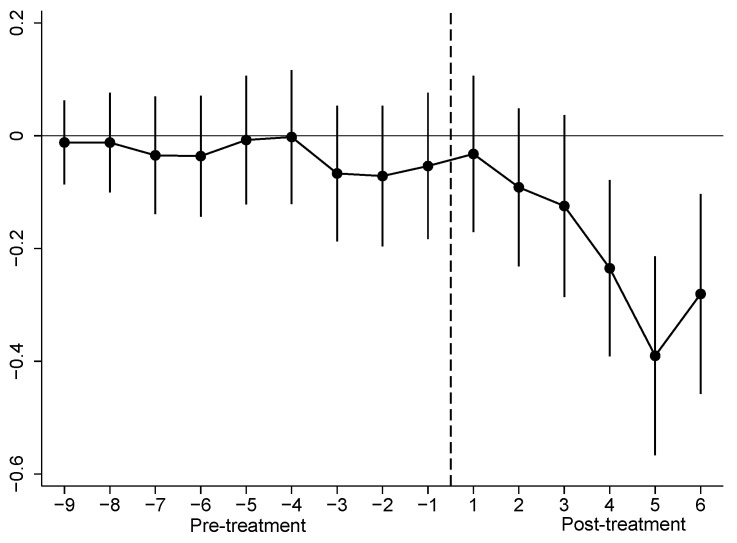
Pretreatment trend and annual treatment effects.

**Figure 2 ijerph-20-00962-f002:**
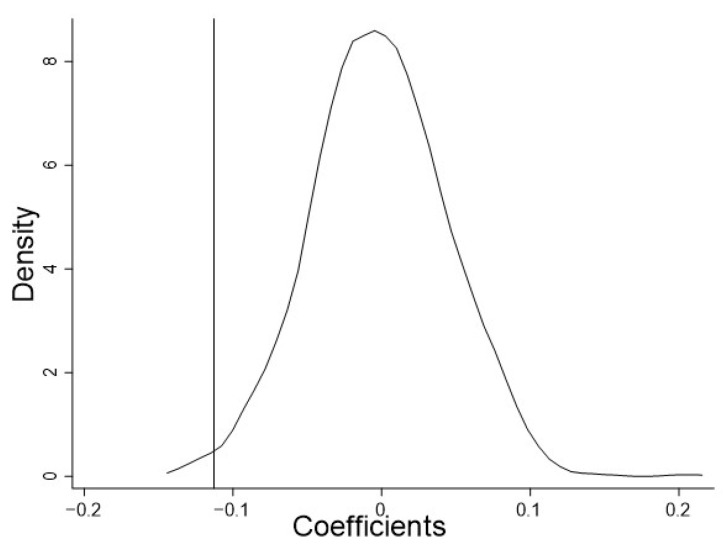
Kernel density of {*β^random^*}.

**Figure 3 ijerph-20-00962-f003:**
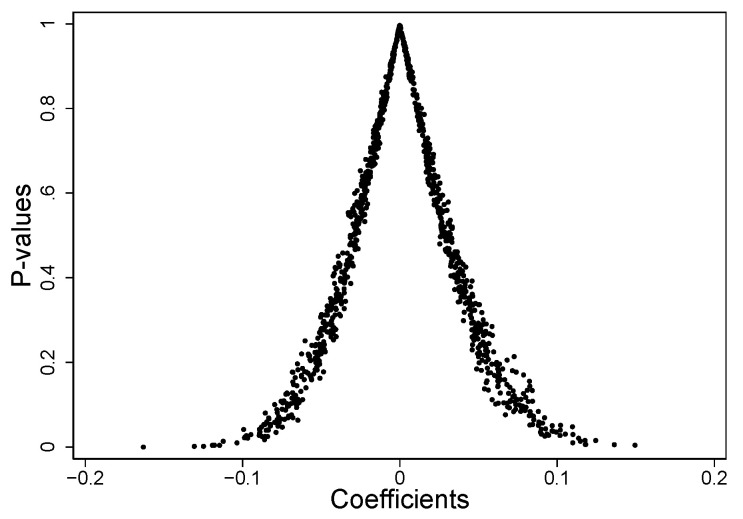
Scatter plot of {*p-value*} versus {*β^random^*}.

**Table 1 ijerph-20-00962-t001:** Variables’ definitions and descriptive statistics.

Variables	Variable Definitions	Obs	Mean	SD
*CEI*	Logarithm of per capita CO_2_ emissions	3976	−10.122	0.817
*BBC*	Equals 1 for “Broadband China” pilot cities in year t, otherwise 0.	3976	0.137	0.343
*Pgdp*	Logarithm of per capita GDP	3976	10.311	0.757
*Pgdp2*	Logarithm of the square of per capita GDP	3976	106.894	15.493
*Pop*	Logarithm of total population	3976	5.864	0.693
*Pub*	Public expenditure on science and technology per public expenditure	3976	0.011	0.014
*Trans*	Paved road area per capita	3976	4.325	6.386
*Hum*	Number of college students per 10,000 population	3976	155.265	187.780
*Fdi*	Foreign direct investment per GDP	3976	0.004	0.016
*Win*	Average daily temperature in January	3976	0.785	8.871
*Sum*	Average daily temperature in July	3976	26.147	2.802

**Table 2 ijerph-20-00962-t002:** The impact of information infrastructure on carbon emission intensity.

	(1)	(2)	(3)	(4)
	TW-FE	TW-FE	TW-FE	IV-2SLS
*BBC*	−0.1611 *** (0.0224)	−0.1139 *** (−0.0223)	−0.1139 ** (−0.0456)	−0.8581 *** (0.2168)
*Pgdp*		0.3853 ****** (0.1601)	0.3853 (0.3041)	−0.3575 (0.5461)
*Pgdp2*		−0.0366 ******* (0.0084)	−03666 ******* (0.0179)	0.0019 (0.0267)
*Pop*		−0.3667 ******* (0.1004)	−0.3667 ******* (0.1630)	0.1700 ***** (0.0941)
*Pub*		−1.4148 ****** (0.6839)	−1.4148 (1.2479)	−1.0457 (1.2261)
*Trans*		−14.0602 (13.6695)	−14.0602 (15.2202)	12.0644 (20.0203)
*Hum*		−0.1974 (1.3715)	−0.1974 (4.1474)	1.9600 (1.9923)
*Fdi*		−0.2831 (0.3816)	−0.2831 (0.3504)	−0.0170 (0.3460)
*Win*		−0.0133 ****** (0.0055)	−0.0133 ****** (0.0058)	−0.0186 ******* (0.0066)
*Sum*		0.0243 ******* (0.0072)	0.0243 ******* (0.0078)	0.0253 ******* (0.0077)
*Rdls*				−0.9228 ******* (0.1365)
Cragg–Donald Wald F				72.71
City fixed effects	YES	YES	YES	YES
Year fixed effects	YES	YES	YES	YES
Obs	3976	3976	3976	3976
R-within	0.4745	0.5277	0.5277	0.5166

Notes: Figures in parentheses are city cluster standard errors; *, **, and *** denote statistical significance levels of 10%, 5%, and 1%, respectively.

**Table 3 ijerph-20-00962-t003:** Robustness test I.

	(1)	(2)	(3)	(4)
	Spatial Spillover		Pre-Existing Trends	
	Contiguity	Inverse Distance	North–South	Provincial Capital
*BBC*	−0.1179 *** (0.0453)	−0.1153 ** (0.0453)	−0.1115 ** (0.0452)	−0.0895 * (−3.36)
*W_DID*	0.0427 *** (0.0198)	0.7054 (0.5995)		
Nor_*T*			0.0135 * (0.0073)	0.0150 ** (0.0073)
Cap_T				−0.0266 *** (0.0090)
Control variables	YES	YES	YES	YES
City fixed effects	YES	YES	YES	YES
Year fixed effects	YES	YES	YES	YES
Obs	3976	3976	3976	3976
R-within	0.5287	0.5287	0.5308	0.5350

Notes: Figures in parentheses are city cluster standard errors; *, **, and *** denote statistical significance levels of 10%, 5%, and 1%, respectively.

**Table 4 ijerph-20-00962-t004:** Robustness test II.

	(1)	(2)	(3)	(4)
	Measurement		Pre-Existing Trends	
	Carbon Emission Per Capita		Common Support	Weighted Regression
*BBC*	−0.1397 *** (0.0443)	−0.0879 ** (0.0427)	−0.1120 ** (0.0464)	−0.1128 * (0.0655)
Control variables	NO	YES	YES	YES
City fixed effects	YES	YES	YES	YES
Year fixed effects	YES	YES	YES	YES
Obs	3976	3976	3620	2242
R-within	0.2988	0.3247	0.5678	0.8545

Notes: Figures in parentheses are city cluster standard errors; *, **, and *** denote statistical significance levels of 10%, 5%, and 1%, respectively.

**Table 5 ijerph-20-00962-t005:** Robustness test III.

	(1)	(2)	(3)	(4)	(5)	(6)
	Difference GMM			System GMM		
*BBC*	−0.0667 *** (0.0200)	−0.0647 *** (0.0190)	−0.0681 *** (0.0189)	−0.0453 ** (0.0226)	−0.0428 ** (0.0212)	−0.0469 ** (0.0213)
*L1.CEI*	0.4840 *** (0.0478)	0.4480 *** (0.0483)	0.4029 *** (0.0505)	0.5858 *** (0.0525)	0.5692 *** (0.0550)	0.5362 *** (0.0547)
*L2.CEI*		0.0689 * (0.0358)	0.0851 ** (0.0391)		0.0471 (0.0349)	0.0711 * (0.0411)
*L3.CEI*			−0.0544 (0.0436)			−0.0560 (0.0452)
Control variables	Yes	Yes	Yes	Yes	Yes	Yes
City fixed effects	/	/	/	/	/	/
Year fixed effects	Yes	Yes	Yes	Yes	Yes	Yes
Obs	3408	3124	2840	3692	3408	3124

Notes: Figures in parentheses are city cluster standard errors; *, **, and *** denote statistical significance levels of 10%, 5%, and 1%, respectively.

**Table 6 ijerph-20-00962-t006:** Information infrastructure and service industry and innovation.

	(1)	(2)	(3)		(4)	(5)
	Service Industry			Innovation		
	Sid1	Sid2	Inn1		Inn2	Inn3
*BBC*	0.7126 * (0.3888)	0.1741 ** (0.0798)	1.9088 ** (0.8930)		0.5075 *** (0.1806)	1.4703 ** (0.6114)
Control variables	YES	YES	YES		YES	YES
City fixed effects	YES	YES	YES		YES	YES
Year fixed effects	YES	YES	YES		YES	YES
Obs	3976	3976	3976		3976	3976
R-within	0.6558	0.1220	0.4739		0.4452	0.4771

Notes: Figures in parentheses are city cluster standard errors; *****, ******, and ******* denote statistical significance levels of 10%, 5%, and 1%, respectively.

**Table 7 ijerph-20-00962-t007:** Information infrastructure and low-carbon lifestyle.

	(1)	(2)	(3)	(4)	(5)
	Low-Carbon Life				
	RCEI	ECEI	GCEI	HCEI	TCEI
*BBC*	−0.052 *** (0.0132)	−0.022 *** (0.0066)	−0.004 * (0.0020)	−0.003 (0.0020)	−0.024 * (0.0076)
Control variables	YES	YES	YES	YES	YES
City fixed effects	YES	YES	YES	YES	YES
Year Fixed effects	YES	YES	YES	YES	YES
Obs	3976	3976	3976	3976	3976
R-within	0.3380	0.1433	0.0399	0.1837	0.3795

Notes: Figures in parentheses are city cluster standard errors; * and *** denote statistical significance levels of 10% and 1%, respectively.

**Table 8 ijerph-20-00962-t008:** Population size and administration rank.

	(1)	(2)	(3)	(4)
	Population Size		Administration Rank	
	High Level	Low Level	High Level	Low Level
*BBC*	−0.1252 ** (0.0508)	−0.0952 (0.0722)	−0.1744 *** (0.0576)	−0.0913 * (0.0527)
Control variables	YES	YES	YES	YES
City fixed effects	YES	YES	YES	YES
Year fixed effects	YES	YES	YES	YES
Obs	3226	2852	2643	3435
R-within	0.5333	0.5035	0.5147	0.5231

Notes: Figures in parentheses are city cluster standard errors; *, **, and *** denote statistical significance levels of 10%, 5%, and 1%, respectively.

**Table 9 ijerph-20-00962-t009:** Human capital and informatization base.

	(1)	(2)	(3)	(4)
	Human Capital		Informatization Base	
	High Level	Low Level	High Level	Low Level
*BBC*	−0.1371 *** (0.0499)	−0.1185 * (0.0710)	−0.1901 ** (0.0477)	−0.0938 (0.0576)
Control variables	YES	YES	YES	YES
City fixed effects	YES	YES	YES	YES
Year fixed effects	YES	YES	YES	YES
Obs	3345	2733	2796	3282
R-within	0.5220	0.5140	0.5176	0.5236

Notes: Figures in parentheses are city cluster standard errors; *, **, and *** denote statistical significance levels of 10%, 5%, and 1% respectively.

**Table 10 ijerph-20-00962-t010:** Marketization and industrialization.

	(1)	(2)	(3)	(4)
	Marketization		Industrialization	
	High Level	Low Level	High Level	Low Level
*BBC*	−0.1282 *** (0.0516)	−0.0966 (0.0738)	−0.1412 *** (0.0522)	−0.0786 (0.0624)
Control variables	YES	YES	YES	YES
City fixed effects	YES	YES	YES	YES
Year fixed effects	YES	YES	YES	YES
Obs	3317	2758	3128	2947
R-within	0.5132	0.5316	0.5278	0.5130

Notes: Figures in parentheses are city cluster standard errors; *** denotes statistical significance levels of 1%.

## Data Availability

Data and materials are available from the authors upon request.

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
