# Peer review of "Does Information Infrastructure Promote Low-Carbon Development? Evidence from the “Broadband China” Pilot Policy"

_ijerph, 2023, doi:10.3390/ijerph20020962_

Round 1
Reviewer 1 Report
This is an interesting paper. Here are my comments in no particular order.
1. Add second-generation panel unit-root test.
2. Why do you use a static panel model instead of a dynamic panel, considering CEI can be persistent?
3. The authors use the relief degree of the land surface (Rdls) as the instrument variable. The endogeneity here is a potential two-way causality between the CEI and BBC. Thus, the authors assume CEI will not directly impact Rdls. Considering that mining and other manufactured pollution-relevant operation occasionally change the topography, I am not fully convinced by this argument. The instrument's validity is simply explained as "First, Rdls, as a geographical indicator, is strongly exogenous." The authors need to discuss more about it.
4. Explain how you choose the control variables. If the selection is following the literatre, please also cite the papers
Author Response
Thanks a lot for having reviewed our manuscript, we appreciate your comments and suggestions for this study. After receiving your review report, based on the comments, we carefully reviewed our manuscript and revised some of its contents. The uploaded document contains the reply to each of your comments and the explanation of the revised content in the new manuscript

Reviewer 2 Report
The authors focused on the impact of information infrastructure on CO2 emissions through the "Broadband China pilot policy". Analyzing the factors affecting CO2 emissions is particularly important for global climate change and sustainable development, especially for a country like China. I congratulate the authors on their choice of topic since the development of the information infrastructure, and spread of information technology is known to have several positive effects, but its impact on a sustainable environment is less studied.
The authors have sought to explore how information infrastructure can shape low-carbon development through a systematic literature review in the 2.2 chapter covering three aspects. First, it would be worthwhile to extend the review by presenting the studies that the researchers have carried out in this area, the methodology they have applied to conduct their studies and the conclusions they have reached. After all, the authors have thoroughly explored the study's theoretical background, but little is known about empirical research on a similar topic to the manuscript, written on the same subject.
At the same time, filling the gap in the empirical literature review also allows the authors to highlight what their study reveals new findings and what gap it fills in the field of research.
The authors have included several indicators as control variables in the model, such as the number of college students per 10,000 population as a measure of human capital. In fact, the indicator appears to be an enrollment rate that does not adequately measure the stock of human capital in the areas included in the study. It would be worthwhile to choose a new indicator or justify why the authors included this variable in their study. Similarly, it needs to justify the inclusion of the variable paved road area per capita for the transportation infrastructure in the model. At the same time, in subsection 6.2, another variable is mentioned for measuring human capital: "employees of information industry per 10,000 population".
In addition to the detailed description and justification of the methodology, it is necessary to provide the interpretation/meaning of the elements of the examined models (equations), such as for equations (7)-(9).
The added value of the study is increased by the implementation of group regression for the group of low and high-level cities separated based on the mean and median.
I recommend that the authors outline further research opportunities at the end of the study.
Author Response

(The authors gave the same response as above.)

Round 2
Reviewer 1 Report
I really enjoyed reading the revised manuscript. I found it more intuitive and streamlined. Thank you also for your reply.